# A Retrospective Study of 10 Patients Exhibiting the “Pseudo Wartenberg Sign”

**DOI:** 10.3390/neurolint17070097

**Published:** 2025-06-20

**Authors:** Lisa B. E. Shields, Vasudeva G. Iyer, Yi Ping Zhang, Christopher B. Shields

**Affiliations:** 1Norton Neuroscience Institute, Norton Healthcare, Louisville, KY 40202, USA; lbes@earthlink.net (L.B.E.S.); yipingzhang50@gmail.com (Y.P.Z.); 2Neurodiagnostic Center of Louisville, Louisville, KY 40245, USA; pavaiyer@gmail.com

**Keywords:** ulnar neuropathy, Wartenberg sign, pseudo Wartenberg sign, abducted little finger, electrodiagnostic study, ultrasonography

## Abstract

Background/Objectives: The Wartenberg sign is a diagnostic feature of ulnar nerve neuropathy. It results from unbalanced activity of the abductor digiti minimi (ADM) and extensor digiti minimi (EDM) muscles secondary to weakness of the third palmar interosseous muscle. Rarely, this sign may occur in the absence of an underlying ulnar neuropathy, which we refer to as the “pseudo Wartenberg sign” (PWS). Methods: This is a retrospective review of 10 patients manifesting an inability to adduct the little finger towards the ring finger with no evidence of an ulnar neuropathy. We describe the clinical and electrodiagnostic (EDX) findings in these patients and discuss the pathophysiologic basis of PWS. Results: The most common cause was an injury in five (50.0%) patients: avulsion of the third volar interosseous muscle in two (20.0%), contracture of the ADM muscle in one (10.0%), and trauma-related dystonia in two (20.0%). The most frequent mechanism of PWS was focal dystonia of specific hand muscles in seven (70.0%) patients. Needle electromyography (EMG) demonstrated no denervation changes in ulnar nerve-innervated hand muscles; the motor and sensory conduction was normal in the ulnar nerve in all patients. Four (40.0%) patients underwent ultrasound studies, with a hyperechoic, avulsed third volar interosseous muscle in one, a hyperechoic and atrophic ADM muscle in one, normal hypothenar and extensor muscles in one, and a normal hypothenar muscle in one. Conclusions: Neurologists, neurosurgeons, and hand and orthopedic surgeons should be aware of the rare cases in which the inability to adduct the little finger may occur in the absence of ulnar neuropathy and look for other causes like avulsion of the third palmar interosseus muscle or focal hand dystonia.

## 1. Introduction

In 1939 Dr. Robert Wartenberg described a sign (the “Wartenberg sign”) that “consists of a position of abduction assumed by the little finger” that “resembled a paralytic contracture” [1]. Wartenberg observed that adduction of the little finger is performed by the interosseous muscles, and abduction is performed by the hypothenar muscles, both of which are innervated by the ulnar nerve. The radial nerve innervates both the extensor digiti minimi (EDM) and extensor digitorum communis (EDC) muscles, which also assist in abducting the little finger. Hence, an intact radial nerve may permit unopposed abduction of the little finger if the ulnar nerve is damaged [1,2]. The mechanism involves tearing of the radial stabilizers of the little finger (junctura tendinum from the EDC muscle, radial sagittal band, and radial collateral ligament) and ulnar displacement of the tendon of the EDM muscle [3]. While several descriptions of the extensor mechanism to the little finger have been reported, the most common is a single slip of the extensor digitorum communis (EDC) muscle and two slips of the extensor digiti minimi (EDM) muscle inserting into the extensor hood [4]. The EDM muscle abducts and hyperextends the little finger, which results in the Wartenberg sign. Wartenberg noted that the position of abduction of the little finger is more pronounced with extension of the fingers at the proximal joints than flexion and that the ring finger may also adopt an abduction position [2].

While the abducted little finger (Wartenberg sign) signifies an underlying ulnar nerve palsy [5], there are a few situations where the sign may be present in the absence of an ulnar nerve neuropathy. This phenomenon, the “pseudo Wartenberg sign” (PWS), can occur in the following situations: (1) avulsion/rupture of the third volar interosseous muscle; (2) contracture and shortening of the abductor digiti minimi (ADM) muscle; (3) dystonia of the ADM/EDM muscles; or (4) fusion of the metacarpophalangeal (MP) joint of the little finger in the abducted position (Figure 1). A few reports have investigated abduction of the little finger unassociated with an ulnar nerve palsy [3,5,6,7,8,9,10,11,12,13,14]. The majority of these cases were traumatic in nature and involved injury of the third volar interosseous muscle, the chief adductor of the little finger.

In this report, we describe 10 patients with PWS with a focus on etiological factors. We describe the probable mechanisms associated with an abducted little finger in the absence of an ulnar nerve palsy.

## 2. Materials and Methods

Our American Association of Neuromuscular & Electrodiagnostic Medicine (AANEM)-accredited Neurodiagnostic Center evaluates approximately 1000 patients each year who are referred for electrodiagnostic (EDX) studies mostly by hand surgeons and neurosurgeons. Under an Institutional Review Board (IRB)-approved protocol, we performed a 6-year (20 March 2019–9 May 2025) retrospective analysis of patients referred for EDX studies who exhibited an inability to adduct the little finger. The EDX protocol included both the nerve conduction and needle EMG studies.

### 2.1. Inclusion and Exclusion Criteria

Inclusion criteria were as follows: (1) patients exhibiting a tendency for the little finger to assume an abducted position and (2) absence of ulnar nerve neuropathy by EDX studies. Patient were excluded if they had weakness of the intrinsic hand muscles caused by (1) ulnar nerve neuropathy, (2) brachial plexopathy, and (3) C8-T1 radiculopathy/neuronopathy. 

Several metrics were collected, including the patients’ gender and age, laterality (left/right), and duration of symptoms, as well as a clinical examination with a particular focus on the motor assessment of the symptomatic hand and mobility of the MP and interphalangeal joints of the fingers.

Ulnar nerve motor conduction velocity across the elbow, forearm, wrist, and palm was determined using the standard protocol in our lab [15]. The study was performed by placing a recording electrode over the ADM muscle as well as the first dorsal interosseous (FDI) muscle. Sensory conduction was also studied with the recording electrode over the digital branch of the ring and little fingers. Needle EMG of the upper extremity muscles included the ADM, FDI, flexor carpi ulnaris [FCU], EDC, EDM, and abductor pollicis brevis (APB). Ultrasound (US) studies were conducted in three patients, focusing on the hypothenar and interosseous muscles.

### 2.2. Institutional Review Board Approval of Research

Informed consent was obtained from all patients. The WCG IRB determined that our study was exempt under 45 CFR 46.104(d)(4). The IRB number is 20251775. The Ethic Approval Code is 05162025. The date of IRB approval was 16 May 2025.

## 3. Results

### 3.1. Demographics

A total of 10 patients had a PWS based on clinical findings and a lack of ulnar nerve neuropathy by EDX studies (Table 1). The mean age was 48.7 years (range: 15–71 years), and the majority (6 [60.0%]) of patients were female. PWS was more common on the right side (6 [60.0%]). Nine (90.0%) patients were right-hand dominant, and one (10.0%) was ambidextrous. The symptomatic side corresponded to hand dominance in seven (70.0%) patients. Five (50.0%) patients had symptoms for 6 months or less, while five (50.0%) experienced symptoms over a longer duration (since childhood in one patient).

### 3.2. Clinical Features

All patients showed an abducted posture of the little finger at rest with difficulty adducting the little finger towards the ring finger. Patient #1 had significant swelling of the fourth and fifth MP joints soon after an injury (Figure 2A). At the time of examination 8 months later, the joints had full mobility with weak adduction of the little finger (Figure 2B) and normal EDX studies. US study findings were consistent with avulsion of the third volar interosseous muscle (Figure 3). Patient #3 also developed a similar picture following a laceration injury between the ring and little fingers. Patient #7 sustained an injury to the wrist in childhood and had difficulty adducting the little finger; an EDX study of the ulnar nerve was normal. A US study suggested contracture of the ADM muscle (Figure 4). Patient #4 sustained an injury to the hand. When evaluated 2 years later, muscle strength was normal, but the little finger showed a tendency to stay abducted (Figure 5). There were features of complex regional pain syndrome (CRPS), and the possibility of dystonia of the ADM muscle as the cause of PWS was considered. Dystonia of the ADM and/or EDM muscles was the underlying cause of PWS in seven (70.0%) cases. The abduction of the little finger was most prominent when the digits were extended at the MP joints and less prominent when the fingers were flexed at the MP joint.

### 3.3. Electrodiagnostic Studies

Needle EMG demonstrated no denervation changes in the ADM, FDI, FCU, and/or EDM muscles in all 10 patients. Motor and sensory conduction of the ulnar nerve was normal in all patients.

### 3.4. Ultrasound Studies

Four (40.0%) of the 10 patients underwent US studies. The US studies demonstrated a hyperechoic and avulsed third volar interosseous muscle in one patient (Patient #1) (Figure 3) and a hyperechoic and little ADM muscle in another (Patient #7) (Figure 4). The hypothenar and extensor muscles appeared normal in one patient (Patient #4), and the hypothenar muscles appeared normal in another (Patient #10).

### 3.5. Etiologies and Mechanisms

Five (50.0%) patients sustained an injury leading to the clinical presentation with PWS, including one in a motor vehicle accident with the little finger caught in the steering wheel and one who experienced an overexertion injury by pulling clothes out of a 60-pound washing machine (Table 1). In two of the five patients, avulsion of the third volar interosseous muscle caused PWS. In one patient, post-traumatic contracture of the ADM muscle led to PWS. In another patient, CRPS complicating the injury may have caused dystonic posturing of the little finger leading to PWS. Five (50.0%) patients did not suffer an injury.

The most common mechanism was dystonia of one or more muscles including the ADM, EDM, or EDC in seven (70.0%) patients. The cause of the focal hand dystonia is unclear in these patients.

## 4. Discussion

A few case reports and small series have reported abduction of the little finger without evidence of an ulnar nerve neuropathy (Table 2) [3,5,6,7,8,9,10,11,12,13,14]. These cases were caused by trauma [3,8,9,12,14], multiple sclerosis with cerebellar lesions [6,13], mild hemiparesis [13], hemiplegic migraine [10], and cord compression above the C6–7 level [11]. The majority of these cases were due to trauma and involved injury of the third volar interosseous muscle. Most of these patients underwent surgical intervention, primarily involving a tendon transfer. Surgery designed to reduce abduction of the little finger may involve an EDM tendon “rerouting” transfer [16]. In this procedure, the EDM tendon is sectioned at its distal insertion and transferred in the wrist through the extensor retinaculum. The tendon is then sutured distally on the radial aspect of the interosseous muscle and to the extensor digitorum tendon of the little finger. The desired goal is for patients to be able to adduct the little finger in extension. Another surgical technique uses a slip of the EDC muscle of the ring finger [17]; the transferred piece may be the central slip or the ulnar slip extended by the connexus intertendineus to the little finger. The extensor mechanism of the little finger is preserved with this technique. However, Lourie et al. recommend that patients initially undergo conservative management with immobilization of the MP joints in flexion and interphalangeal joints in extension [3].

Several situations have been reported where the little finger remains in an abducted posture. In his landmark article reporting the sign of an abducted little finger associated with ulnar nerve palsy, Wartenberg acknowledged that this same sign may occur in cases of cerebellar disease and multiple sclerosis with cerebellar lesions as described by Hoff and Schilder as “little finger phenomenon” in 1928 [2,6]. Abduction of the little finger has been observed in a patient with multiple sclerosis who had pyramidal tract and cerebellar signs [13]. The hyperextended posture of the hand and fingers usually seen with cerebellar lesions may distinguish a cerebellar from a pyramidal tract lesion [13]. Additionally, infants with incomplete myelination of the pyramidal tract may exhibit abduction of the little finger when grasping a crib rail [13]. This resolves with age as myelination progresses.

Another situation where the little finger is abducted during certain movements is known as the “digiti quinti sign” (DQS), occasionally seen in patients with mild hemiparesis [13]. This sign can be elicited by having the patient extend the arms and fingers forward with the palms down. Fingers are normally adducted in this position. On the hemiparetic side, the little finger may be abducted off the ring finger [13]. Alter described how the DQS may be confused with abduction of the little finger (Wartenberg’s sign) associated with an ulnar nerve palsy [13]. This author details the other signs related to an ulnar nerve neuropathy such as sensory loss of the little finger and ulnar half of the ring finger as features that differentiate the two conditions. The DQS has also been reported in patients with hemiplegic migraine ipsilateral to the motor deficits [10]. Vincent described three patients with hemiplegic migraine with a DQS as the sole neurological abnormality [10]. All three had brain MRIs without lesions relating to the hemiparesis, and the DQS was observed interictally. Abduction of the little finger unassociated with ulnar nerve palsy has also been reported in spinal cord compression above the C6–7 level [11]. The finger escape sign is provoked by asking the patient to extend and adduct the fingers and documenting the tendency for the little finger to drift into abduction in less than 30 s [11]. In these situations, it is likely that corticospinal tract involvement may be causing the involuntary abduction of the little finger, similar to the well-known pronator drift. An EMG is a valuable test to distinguish true Wartenberg sign of ulnar nerve palsy from a PWS secondary to different etiologies, including corticospinal tract lesions, cervical radiculopathy/myelopathy, focal hand dystonia, and local trauma [4,9−14,17].

In the present study, five patients sustained an injury causing the PWS; two had avulsion of the volar third interosseous muscle, one had contracture of the ADM muscle, and the other two developed dystonia. In seven patients, PWS was attributed to focal hand dystonia of either the ADM or EDM muscles, the EDC muscle, or a combination of these. Focal hand dystonias are characterized by abnormal, involuntary muscle contractions that manifest as twisting movements and abnormal postures of the hand, wrist, or forearm [18]. They are marked by a lack of reciprocal inhibition (control of muscles around a single joint) and surround inhibition (selective control of individual muscles by simultaneously inhibiting surrounding muscles) [19]. There is an abnormally prolonged muscle firing with an imbalance between excitation and inhibition of neural circuits.

Focal dystonias of the hand are often task-specific, like writer’s cramp and musician’s dystonia [19,20]. Focal non-task-directed dystonia of the little finger has been reported in a patient with ulnar nerve neuropathy at the elbow [20]. These authors noted episodic dystonic adduction and flexion of the little finger, opposite of the posture in PWS. The patient attained complete improvement of the dystonic movement following a cubital tunnel release. Most cases of dystonia are “primary” or “idiopathic,” although they are also known to occur rarely in the context of trauma and nerve entrapment [20]. They tend to be “fixed dystonias” and lead to diagnostic challenges, especially when the clinical feature is the Wartenberg sign. In the context of dystonia manifesting as an abducted and extended small finger, a thorough evaluation of the ulnar nerve with detailed EDX studies is crucial before concluding that the underlying cause is dystonia. In our series, dystonia was unassociated with ulnar neuropathy and involved the ADM and/or EDM/EDC muscles causing the abducted posture of the little finger.

While EDX studies are crucial in differentiating a true Wartenberg sign from PWS, US studies provide valuable information regarding the underlying cause. The protocol should include imaging the ulnar nerve from the wrist to the axilla to look for focal enlargement and altered echogenicity. If the nerve is normal, a more detailed study is needed to visualize the muscles and joints that are relevant to the abducted posture of the small finger, especially the hypothenar and extensor muscles. In cases of focal dystonia causing PWS, US is also useful in detecting the location of muscles for botulinum toxin injections.

### Strengths and Limitations

The strength of the present study is that it features the largest number of patients with PWS, a rare phenomenon. By identifying the presence of PWD and determining its etiology in each patient, the most effective treatment course can be initiated. For dystonic PWS, injection of botulinum toxin into the ADM, EDM, or EDC muscles may be effective. For post-traumatic PWS, the underlying cause can be found by imaging including US, and appropriate treatment can be pursued. Limitations of this study include its retrospective nature and lack of follow-up. US studies were only performed in three cases, and no MR imaging studies were done. Our findings need to be confirmed in a larger series of patients.

## 5. Conclusions

Neurologists, neurosurgeons, and hand and orthopedic surgeons should be aware that a Wartenberg sign does not invariably indicate an underlying ulnar nerve neuropathy, as rarely it may be due to other conditions that may be causing the abduction of the little finger. EDX studies are essential for verifying the absence of ulnar nerve pathology. Imaging studies such as US may elucidate the etiology of the PWS and guide further management.

## Figures and Tables

**Figure 1 neurolint-17-00097-f001:**
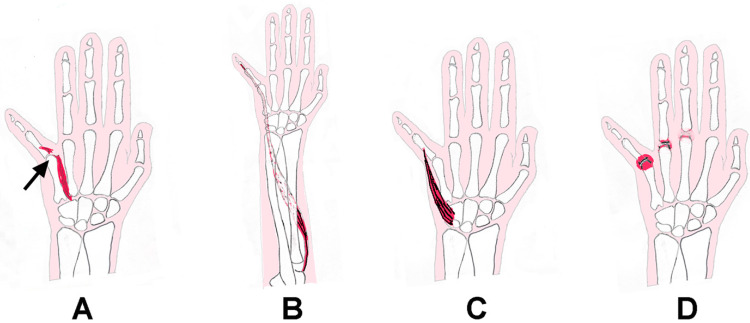
Drawing depicting (**A**) palmar view of the right hand showing avulsion/rupture of the third volar interosseous muscle (arrow), (**B**) palmar view of the right hand/forearm depicting contracture or dystonia of extensor digiti minimi, (**C**) palmar view of the right hand showing contracture or dystonia of abductor digits minimi, and (**D**) palmar view of the right hand depicting fusion of the MP joint of the little finger.

**Figure 2 neurolint-17-00097-f002:**
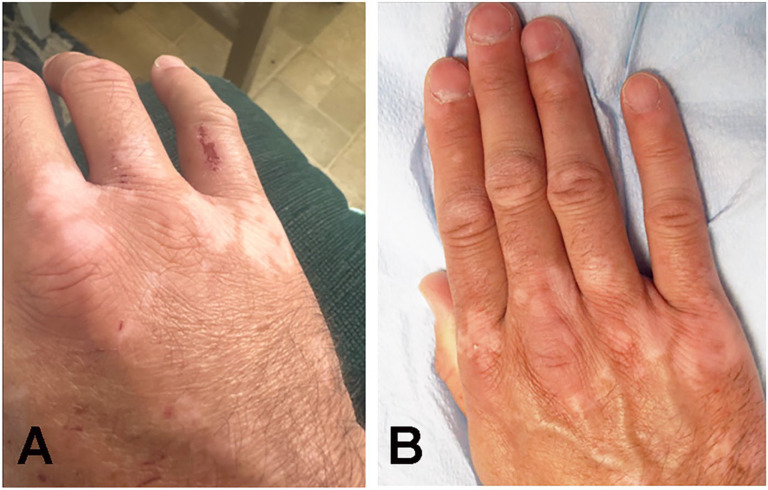
Patient #1: (**A**) Soon after the injury edema was noted around the third, fourth, and fifth metacarpophalangeal joints. (**B**) Eight months after the injury, the patient was unable to completely adduct the little finger.

**Figure 3 neurolint-17-00097-f003:**
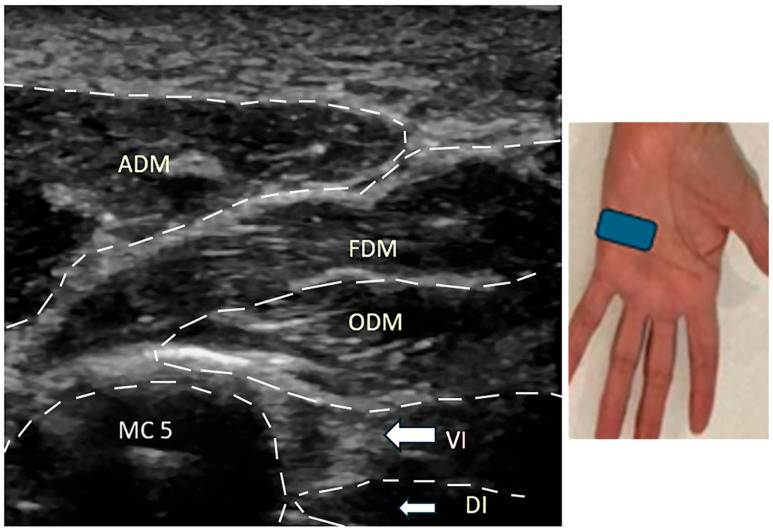
Patient #1: Ultrasound study of the short axis view at the distal hypothenar area showing hyperechoic appearance of volar interosseus muscle. The blue box shows the location of the ultrasound probe. ADM: abductor digiti minimi. FDM: flexor digiti minimi. ODM: opponens digiti minimi. VI: volar interosseous muscle. DI: dorsal interosseous muscle. MC 5: metacarpal bone of digit 5.

**Figure 4 neurolint-17-00097-f004:**
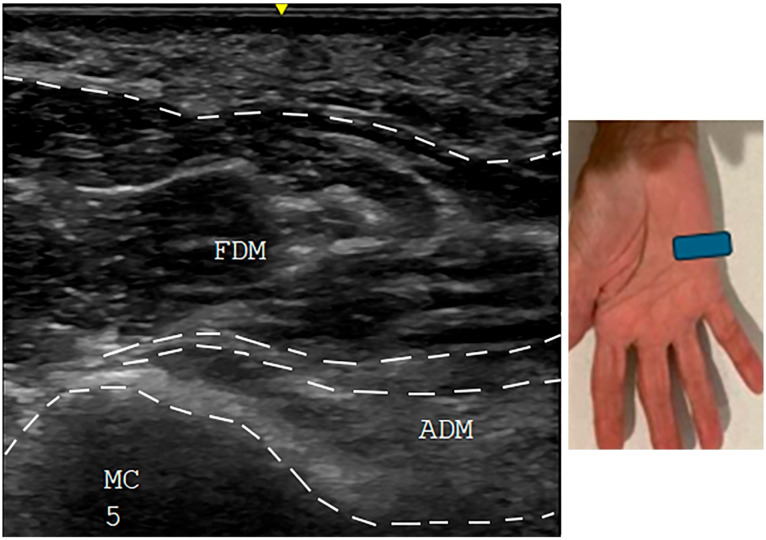
Patient #7: Ultrasound study of the short axis view of left hypothenar muscles showing a hyperechoic ADM muscle. The blue box shows the location of the ultrasound probe. FDM: flexor digiti minimi. ADM: abductor digiti minimi. V M: fifth metacarpal.

**Figure 5 neurolint-17-00097-f005:**
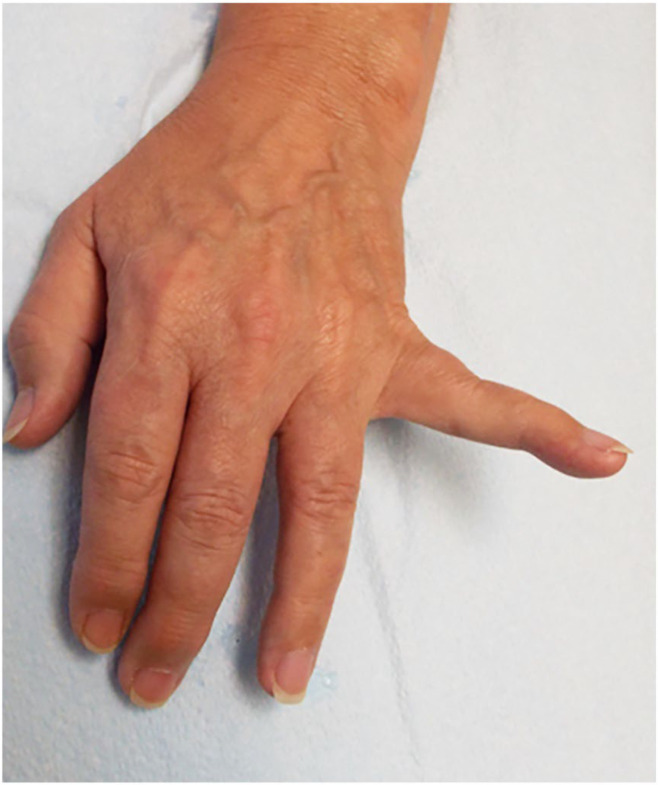
Patient #4: Abducted and extended little finger from dystonia.

**Table 1 neurolint-17-00097-t001:** Clinical characteristics of patients with pseudo Wartenberg sign.

Case Number	Age (Years)/ Gender	Cause	Mechanism	Duration of Symptoms	Side	Physical Examination
1	48/M	MVA: little finger caught in steering wheel	Injury: avulsion of third volar interosseous	8 months	Right	Little finger stays abducted; weakness of adduction of little finger towards ring finger
2	18/M	No injury	ADM/EDM dystonia	2 weeks	Right	Little finger stays abducted, more when fingers are fully extended
3	57/M	Laceration: between digits 4 and 5	Injury: avulsion of volar third interosseous muscle	5 months	Left	Little finger stays abducted; weakness of adduction of little finger towards ring finger
4	54/F	Pulling clothes out of 60-pound washing machine	Injury: ADM/EDC dystonia secondary to CRPS	2 years	Left	Little finger stays abducted; tenderness left hypothenar area/ little finger/ulnar palm; erythema/allodynia/ hyperalgesia over hypothenar area
5	54/F	Injury mostly of digit 3	ADM/EDM dystonia	10 months	Right	Little finger stays abducted; weakness of adduction of little finger towards ring finger; able to voluntarily adduct little finger when the digits are flexed at the MP joints
6	71/F	No injury	ADM/EDM dystonia	6 months	Left	Little finger stays abducted; weakness of adduction of little finger towards ring finger
7	61/M	Injury of left wrist	Injury: contracture of ADM	childhood	Left	Little finger stays abductued; weakness of adduction and abduction of little and ring fingers
8	57/F	No injury	ADM/EDM dystonia	6 months	Right	Little finger stays abducted; weakness of adduction of little finger towards ring finger when fingers are extended
9	52/F	No injury	ADM/EDM dystonia	1 year	Right	Little finger stays abducted; weakness of adduction of little finger towards ring finger when fingers are extended
10	15/F	No injury	EDM dystonia	4 months	Right	Little finger stays abducted; hyperextension at the MP joint when adduction is attempted

MVA: motor vehicle accident, ADM: abductor digiti minimi, EDM: extensor digiti minimi, EDC: extensor digitorum communis, CRPS: complex regional pain syndrome, MP: metacarpophalangeal.

**Table 2 neurolint-17-00097-t002:** Abduction of the little finger (pseudo Wartenberg sign) caused by trauma in the literature.

Publication	Number of Patients	Cause	Mechanism	Treatment
Freeland et al. [12]	1	Injury: playing ball	Deformity of little finger caused by avulsion of the insertion of the third volar interosseous muscle	Transferring ulnar tendon of EDM to phalangeal attachment of the radial lateral band
Lourie et al. [3]	6	Unspecified trauma	Axially directed force to little finger with hyperabduction	Five patients had 4 weeks of immobilization holding metacarpophalangeal joints in 40° flexion and the interphalangeal joins in extension; three of these patients underwent plication of radial collateral ligament and imbrication of radial sagittal band and junctura tendinum
Yacoubi et al. [9]	1	Unspecified trauma	Avulsion of the insertion of the third volar interosseous muscle	Transfer of the EDM onto the radial side of the EDC
Goto et al. [14]	1	Injury: laceration between ring and little fingers	Third volar interosseous muscle injury	Fourth flexor digitorum superficialis tendon transfer
Morisaki et al. [8]	1	Injury: lifting heavy object with hyperabducted little finger	Rupture of the third volar interosseous muscle	Transfer of the fourth dorsal interosseous muscle to the lateral band of the little finger

EDM: extensor digiti minimi, EDC: extensor digitorum communis, EDX: electrodiagnostic.

## Data Availability

All of the data for this study is included in the current article.

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
