# Peer review of "A Retrospective Study of 10 Patients Exhibiting the “Pseudo Wartenberg Sign”"

_2035-8377, 2025, doi:10.3390/neurolint17070097_

Round 1

Reviewer 1 Report (Previous Reviewer 2)

Comments and Suggestions for Authors

All the comments have been answered.

Reviewer 2 Report (Previous Reviewer 1)

Comments and Suggestions for Authors

Thank you to the author for making the necessary corrections.

This manuscript is a resubmission of an earlier submission. The following is a list of the peer review reports and author responses from that submission.

Round 1

Reviewer 1 Report

Comments and Suggestions for Authors

The manuscript entitled “ A Retrospective Study of 10 Patients Exhibiting the 'Pseudo Wartenberg Sign ” submitted to Neurology International is an article about the Pseudo Wartenberg Sign. It discusses the several causes of the said conditions and highlights valuable insights into the inability to adduct the little finger in the absence of ulnar neuropathy.

The authors may consider the minor comments and suggestions to enhance the clarity of the manuscript.

  1. Abstract: Line 39, Page 1, "EMG" has been used as an acronym. Its larger form could be added.
  2. Material and Methods: Line 80, Page 3, "AANEM" has been used as an acronym. Its expanded form could be added.
  3. Results: The tables are organized lucidly, facilitating comparisons across various equations.
    Table 1, Page 3, The second column representing the age and gender doesn’t have units for Age. It could be added as (Age (Years)

In succinct, the authors provided a systematic analysis of the case history of the 10 patients) said rare condition.

This article highlights the importance of knowledge of the Pseudo Wartenberg Sign. The manuscript has been presented lucidly with an organized format. Understanding these issues is crucial for improving outcomes and empowering individuals affected by the Pseudo Wartenberg Sign.

Reviewer 2 Report

Comments and Suggestions for Authors

This is a very interesting and well-written work. Wartenberg's sign is rare and consists in the inability to adduct the little finger. We encounter such cases mainly in neurophysiological laboratories when patients are referred for ulnar neuropathy. In these cases, it is important to know the alternative diagnosis in order to avoid unnecessary surgery.

I would just like to make a few suggestions to the authors.

In Table 1, MVA is not explained in the footnotes.

I believe Table 2 can be omitted, since the EDX studies are identical in all cases and the U/S studies were only performed in 3/10 patients. The authors can only describe the U/S findings of the 3 patients.

This paper posits that dystonia is the most common underlying cause of Wartenberg's sign, and that it is likely a diagnosis of exclusion. The authors are recommended to dedicate a greater proportion of the paper to the description of the clinical features of dystonia and the challenges associated with arriving at a definitive diagnosis. In cases of injury, ultrasonography (US) has been demonstrated to be a valuable diagnostic tool, as it can provide a visualisation of the injured muscle (ADM)   I suggest that greater emphasis ought to be placed on the diagnostic value of ultrasound by the authors of the paper.